# One-Step Preparation of S, N Co-Doped Carbon Quantum Dots for the Highly Sensitive and Simple Detection of Methotrexate

**DOI:** 10.3390/molecules27072118

**Published:** 2022-03-25

**Authors:** Xiaoyi Wei, Xiaojing Si, Mei Han, Chen Bai

**Affiliations:** Department of Food Science, Shanghai Business School, Shanghai 200235, China; xiaoyi_wei@163.com (X.W.); hanmei.918521@163.com (M.H.); baichen_sh@163.com (C.B.)

**Keywords:** S, N co-doped carbon quantum dots, fluorescence, methotrexate, inner filter effect

## Abstract

(1) Background: Carbon quantum dots (CQDs) are a new class of carbon nanomaterials with favorable features, such as tunable luminescence, unique optical properties, water solubility, and lack of cytotoxicity; they are readily applied in biomedicine. (2) Methods: S, N co-doped CQDs were prepared to develop a highly selective and sensitive fluorescence technique for the detection of methotrexate (MTX). For this purpose, citric acid and thiourea were used as C, N, and S sources in a single-step hydrothermal process to prepare the S, N co-doped CQDs, which displayed remarkable fluorescence properties. (3) Results: Two optimal emissions were observed at the excitation/emission wavelengths of 320/425 nm, respectively. The two emissions were significantly quenched in the presence of MTX. Under optimal conditions, MTX was detected in the linear concentration range of 1–300 μmol/L, with the detection limit of 0.33 μmol/L. The sensing mechanism was due to the fact that the effect of the inner filter on MTX and S, N-CQDs causes fluorescence quenching. The contents of MTX in real medicine samples were evaluated with acceptable recoveries of 98–101%. (4) Conclusions: This approach has great potential for detecting MTX in pharmaceutical analysis.

## 1. Introduction

Methotrexate (MTX) is a folic-acid-type antitumor agent that is commonly used in high doses in the treatment of lymphoma, acute lymphoblastic leukemia, chorionic epithelial carcinoma, and other types of malignant tumors, as well as rheumatoid arthritis [1]. MTX is used to prevent the transferring of folic acid into folinic acid with physiological activity through inhibiting the activity of dihydrofolate, which can reduce the synthesis of DNA and protein, then block the synthesis of tumor cells [2]. It is not only effective against tumor cells, but also against healthy cells and tissues, resulting in a variety of severe adverse effects that are extremely toxic, strong, and long-lasting, most notably bone marrow suppression [3,4]. MTX in the human body is mainly excreted by the kidneys as the original drug, and a small part is excreted as 7-hydroxymethotrexate by liver metabolism. However, in an acidic environment, 7-hydroxymethotrexate precipitates in the proximal tubule of the kidney, causing kidney damage [5]. Furthermore, MTX may be found in hospital wastewater, which flows into the environment, destroys the ecosystem, and poses a risk to ecological safety and human health [6]. Therefore, it is imperative to determine the amount of MTX in drug and in vivo analyses.

There are several methods available for determining MTX, such as surface-enhanced Raman scattering (SPRS) [1], surface plasmon resonance (SPR) [2], high-performance liquid chromatography (HPLC) [7,8,9], electroanalysis [10,11,12], and immunoassays [13]. In comparison to the methods mentioned above, fluorescence has gained attention due to its remarkable sensitivity, quick response, easy operation, good biocompatibility, minimal toxicity, and the ability to detect non-destructively [14,15]. Carbon quantum dots (CQDs) are a new class of carbon nanomaterials, quasi-spherical carbon nanoparticles with core–shell structure, and sizes of less than 10 nm, that exhibit outstanding properties such as ease of preparation, non-toxic nature, and excellent luminescence behavior [16,17]. It has recently been demonstrated that doping with a heteroatom, such as sulfur, nitrogen, phosphorus, boron, and fluorine, can improve the quantum yields (QY) of CQDs [18,19]. Doping with different heteroatoms may result in different changes in morphology, composition, and structure, resulting in differential characteristics; it can open a modest band gap in the electrical structure of CQDs, demonstrating their potential for several of the applications listed above [20,21]. Recent studies have shown that N-doping and S-doping are the most commonly used strategies to improve the optical and material properties of CQDs, because the atomic radius of N and the electronegativity of S are similar to those of carbon, and show an improved electrocatalytic activity and a higher QY, thus paving the way for a new generation of smart electrocatalysts, whose activity could be correlated with light emission [22,23,24]. Compared with single-heteroatom doping, co-doping multiple heteroatoms will greatly improve the performance of CQDs due to the synergistic effect between the doped heteroatoms [25,26,27]. Moreover, N, S, and C are essential elements in the organism. Therefore, the study of S and N co-doped CQDs (S, N-CQDs) is very meaningful for CQDs in biomolecule and pharmaceutical analysis. However, there are few reports about the applications of S, N-CQDs for pharmaceutical analysis, and their use to detect MTX [28,29,30].

Therefore, utilizing citric acid as a source of carbon and thiourea as a source of nitrogen, we synthesized novel S, N-CQDs through one-step hydrothermal synthesis. The modern analysis methods were taken for the characterization of microstructure, composition, and optical properties. The experimental results demonstrate that the fluorescence sensor exhibits comparable accuracy, good sensitivity, and may be effectively used to determine the presence of MTX in medicines.

## 2. Results

### 2.1. Morphological and Microstructural Characterization of the Synthesized S, N-CQDs

The morphological properties and microstructure of S, N-CQDs are depicted in Figure 1 using an HRTEM (Figure 1A), a nanoparticle analyzer (Figure 1B), an energy-dispersive X-ray spectrometer (EDS, Figure 1C), and X-ray photoelectron spectroscopy (XPS, Figure 1D) The synthesized S, N-CQDs had a homogeneous particle size distribution with a mean diameter of 7 nm. Their round-shaped morphology is typical for CQDs obtained by the bottom-up approach and corresponds to other researchers’ results [16]. The EDS result showed the C, O, N, and S elemental contents in the synthesized S, N-CQDs were 90.3%, 5.98%, 2.65%, and 1.07%, respectively. The XPS presents four peaks at 169, 285, 400, and 532 eV, corresponding to S 2p, C 1s, N 1s and O 1s, respectively. The findings revealed the successful co-doping of S and N. Therefore, there are C-C, C-S, C-O, C=O, C-N-C and C-S-C units in S, N-CQDs [31].

The IR spectrum of the synthesized S, N-CQDs is depicted in Figure 2A. The broad absorption bands at 3184 and 3006 cm^−1^ were attributed to the N-H and O-H stretching vibrations, respectively, demonstrating the presence of a large number of amino and hydroxyl groups on the surface of S, N-CQDs, which contributed to their hydrophilic properties. The strong absorption peak at 1650 cm^−1^ was a result of the C=O stretching band, and the peaks at 1556 cm^−1^ and 1405 cm^−1^ indicated the asymmetric and symmetric COO^−^ stretching of carboxylates, respectively. Furthermore, the absorption peak at 1194 cm^−1^ was found to correlate to the stretching vibration of C=S [32]. This observation is consistent with the XPS results. On the basis of the experimental results, we conclude that S, N-CQDs can interact between thiourea and citric acid under high temperature and pressure through the interactions. In the synthesis of S, N-CQDs, the amine group in thiourea can react with hydroxyl groups in citric acid to form hydrogen bond, the amine group can react with carboxyl groups in citric acid to form ionic bond, and the amine group can also react with carbonyl groups to form covalent bonds. S, N-CQDs have an abundance of highly electronegative NH_2_ functional groups on their surface. This could be related to hydrogen bond formation with a significant number of hydroxyl groups in the citric acid molecule, which increases the efficiency of fluorescence quenching [33]. Additionally, XRD was utilized to characterize the crystal structure of S, N-CQDs, as illustrated in Figure 2B. The diffraction peak of amorphous carbon was observed at 2θ = 21.46°, and the crystal structure of S, N-CQDs was determined to be amorphous carbon.

### 2.2. Optical Properties of the Synthesized S, N-CQDs

The optical properties of S, N-CQDs were investigated using UV–Vis absorption spectra and fluorescence measurements. The UV–Vis spectrum exhibited a distinctive absorption peak at 325 nm, which was caused by n-π* transitions between C-N and C=S bonds, as illustrated in Figure 3A. The S, N-CQDs exhibited high fluorescence intensity in the absence of MTX. The optimum wavelengths for S, N-CQDs excitation, and emission were 320 nm and 425 nm, respectively. Meanwhile, the peaks were symmetrical. As shown in a and b of Figure 3A, the UV absorption spectrum and the excitation spectrum of the S, N-CQDs exhibit a strong overlap. According to previous research, the quenching mechanisms for MTX to S, N-CQDs may be related to the inner filter effect (IFE) and electron transfer mechanism. p-p stacking occurs between the aromatic heterocyclic part of S, N-CQDs and the pteridine ring or phenyl group of MTX. With these specific interactions, N, S co-doped CQDs strongly target methotrexate to serve as a chemosensor for the determination of methotrexate based on IFE [15].

As a result, the presence of MTX can be detected using a considerable change in fluorescence intensity generated by the inner filter effect [34]. Furthermore, when the wavelength of excitation was altered from 280 nm to 370 nm, the wavelength of maximum emission was determined to be 420 nm with a 20 nm Stokes shift, suggesting that the emission wavelength of S, N-CQDs was excitation-dependent, as illustrated in Figure 3B. This phenomenon could be explained by the S, N-CQDs’ size effect. Multiple energy levels result from differences in particle size, number, and location of emission sites [35]. The QY of S, N-CQDs was also determined, which was found to be 10.3%.

### 2.3. Conditions Optimization

To determine the MTX concentration in vivo and in vitro, PBS was utilized as the reaction medium. However, because MTX has a variable quenching influence on the fluorescence intensity of S, N-CQDs at different pH values, the pH was studied. MTX had a significant quenching influence on S, N-CQDs in a 10 mL PBS (pH = 6.5) solution containing 200 μL of S, N-CQDs, and 50 μL of 0.01 mol/L MTX under acidic conditions, as indicated in Figure 4A, and the quenching effect decreased with increasing pH. The S, N-CQDs were pH-sensitive with a broad response region. Additionally, changes in pH had a minimal effect on the intensity of the fluorescence of S, N-CQDs under alkaline conditions. Thus, to facilitate the in vivo study, MTX was measured in a PBS buffer solution with a pH of 6.5.

The higher the quenching efficiency, the better for sensitivity detection, and the different amounts of CQDs produced different quenching efficiencies. In the experiment, different volumes of S, N-CQDs (50–250 μL) were added, mixed with 50 μL of 0.01 mol/L MTX, and the fluorescence intensity was measured. The fluorescence intensity increased gradually as the quantum dot concentration increased, as shown in Figure 4B, but at 200 μL, MTX had a significant quenching effect on CQDs, resulting in a decrease in fluorescence intensity. However, as the quantum dot dosage was increased further, the intensity of the fluorescence slightly increased. As a result, 200 μL S, N-CQDs were determined to be preferred.

### 2.4. Calibration Curve

It is well established that linear range and detection sensitivity are critical parameters for a fluorescence quenching detection system [36]. It is important to observe that the fluorescence intensity falls significantly after MTX adsorption. According to Figure 5, S, N-CQDs undergo fluorescence quenching in the presence of MTX at concentrations ranging from 1 to 300 μmol/L under optimal conditions with a limit of detection (LOD) of 0.33 μmol/L (S/N = 3). Additionally, this result is consistent with the Stern–Volmer-type equation: log (F_0_/F) = 0.0064C_MTX_ + 0.0875, (R^2^ = 0.9945), where F_0_ and F denote the fluorescence emission intensities of MTX with and without the S, N-CQDs solution at a 320 nm excitation wavelength, respectively. In comparison with existing analytical methods for MTX detection, the S, N-CQDs fluorescence analysis approach demonstrated a larger linear range and higher accuracy. Other MTX detection methods previously reported were also compared and are listed in Table 1.

### 2.5. Interferencing Test

To evaluate the selectivity of S, N-CQDs as fluorescent probe for MTX detection, several commonly used external interferents were added to a PBS solution containing 50 μL of 0.01 mol/L MTX according to the experimental protocol under optimal test conditions. The results are shown in the Figure 6. When the relative error is within ±5%, 500-fold dilutions times of Na^+^, K^+^, Ca^2+^, Zn^2+^, 100-fold dilutions of glucose, fructose, sucrose, and 50-fold dilutions of glycine, aspartic acid, phenylalanine do not interfere with the determination. The approach demonstrated a high degree of selectivity and suitability for the quantitative detection of MTX.

### 2.6. Reproducibility and Stability

The produced S, N-CQDs solution was irradiated with a UV light for 60 min and then kept at 4 °C for 7 days. Following that, 200μL S, N-CQDs solution was added to 50 μL 0.01 mol/L MTX standard solution, bringing the total volume to 10 mL of PBS (pH = 6.5). The fluorescence intensity was evaluated after shaking. The average fluorescence intensity decreased to 95.44%, with an RSD of just 0.80%, demonstrating that S, N-CQDs exhibit excellent photostability.

### 2.7. Analytical Application

To demonstrate the method’s viability, MTX tablets were used as the actual samples to be determined in the investigation: 100 μL serum was diluted 100 times with PBS (pH 6.5). After that, 100 μL supernatant containing MTX (1 mg/mL) was added, and the specific experimental results are listed in Table 2. The average labeled quantity was 99%, with an RSD of 1.1%. Furthermore, the standard addition method was used to calculate the experiment’s recovery rate and RSD. Table 3 shows the results of the addition of 80 μL of 0.01 mol/L MTX standard solution to the solution containing the 100 μL MTX supernatant. The average recovery rate was 94.9%, with an RSD of 2.13%. Due to its advantageous properties, such as high sensitivity, low cost, time savings, simple operation, and convenient analysis, the suggested fluorescence sensor provides a promising option for the rapid detection of MTX in associated environmental and biological samples.

## 3. Materials and Methods

### 3.1. Materials

MTX hydrate (>97%) was obtained from Shanghai Titan Technology Co., Ltd. (Shanghai, China). Thiourea (>99%) and citric acid (>99.5% purity) were purchased from Sinopharm Chemical Reagent Co., Ltd. (Shanghai, China). Methotrexate tablets were produced by Shanghai Xinyi Pharmaceutical Co., Ltd. (Shanghai, China). All analytical measurements were performed using ultrapure water (18.25 MΩ cm).

### 3.2. Instrumentation

The lifetimes of fluorescence were determined using a spectrophotometer (Hitachi, F-7000, Tokyo, Japan). The S, N-CQDs microstructure was studied using high-resolution transmission electron microscopy (HRTEM) (Tecnai G2 F20, FEI, Hillsboro, OR, USA) at a 200 kV working voltage. The S, N-CQD sizes were determined using the Zetasizer Nano ZS 90 Zeta potential analyzer (Malvern, Melvin City, UK). Fourier transform infrared (FTIR) spectra of pressed KBr pellets were recorded on an AVATAR-370 spectrometer (Nicolet, Brookfield, WI, USA) throughout a 400–4000 cm^−1^ spectral range. X-ray diffraction (XRD) images were obtained using D8 advance (Bruker, Karlsruhe, Germany). X-ray photoelectron spectrometry (XPS) mages were obtained using K-Alpha spectrometer (Thermo Fisher Scientific, Waltham, MA, USA). The UV–Vis absorption spectra were measured on a UV-2501PC spectrophotometer (Shimadzu, Kyoto, Japan).

### 3.3. Methods

#### 3.3.1. Synthesis of S, N-CQDs

A one-step hydrothermal approach was used to synthesize the S, N-CQDs, according to that previously reported in the literature, with modifications [38]. The citric acid (0.5764 g, 0.003 mmol) and thiourea (0.4567 g, 0.006 mmol) were dissolved in 30 mL ultrapure water, homogenized with ultrasonication for 20 min, then placed in a Teflon autoclave (50 mL) and heated for 12 h at 180 °C. The samples were then brought to room temperature. The resultant mixture was centrifuged for 10 min at 5000 rpm and then purified further via a 0.22 μm filter membrane. Then, the transparent aqueous solution was then subjected to dialysis (Mw = 1000 Da) for 2 days to eliminate the overreacted residue. Dialysis was then performed for the solution to eliminate any remaining precursors or side products. After that, clear and transparent S, N-CQDs were obtained and stored in a refrigerator at 4 °C for future usage.

#### 3.3.2. Measurement of Fluorescence Quantum Yield

The fluorescence QY of S, N-CQDs could be determined and calculated following the method reported earlier [24]. The formula is as follows: QY = QY_R_ × (I/I_R_) × (A_R_/A) × (n^2^/n_R_^2^). Quinine sulfate (QY_R_ = 54%) was a reference, S, N-CQDs quantum yields were Ex = 325 nm. I and IR denoted the integrated intensities of S, N-CQDs, and quinine sulfate, respectively; A and AR denoted the absorbance of S, N-CQDs, and quinine sulfate at 325 nm, respectively; and n denoted the solution’s refractive index, where S, N-CQDs were dissolved in ultrapure water (n = 1.33) and quinine sulfate in 0.1 mol/L H_2_SO_4_ (n_R_ = 1.33). The absorbance value must be less than 0.05.

#### 3.3.3. Pretreatment and Detection of MTX Tablets

Ten MTX tablets were weighed and fully ground, and 10 mg of MTX powder was accurately weighed and transferred to a clean 10 mL volumetric flask. An appropriate amount of 0.1 mol/L NaOH was added to completely dissolve the powder, and then ultrapure water was added to the scale, mixed well, and filtered by a 0.22 μm filter membrane, and the supernatant was used as the real sample’s solution.

The following procedure was used to determine the presence of MTX in real samples: 200 μL S, N-CQDs solution was poured into a series of colorimetric tubes (10 mL); then, 0.01 M MTX solution in different volumes was added to each colorimetric tube in turn, followed by phosphate-buffered solution (PBS) solution (pH 6.5) as the calibration line. After 3 min, the fluorescence intensity was detected at a 360 nm excitation wavelength.

## 4. Conclusions

S, N-CQDs with distinctive emission characteristics were synthesized through a simple hydrothermal approach using thiourea and citric acid. The S, N-CQDs were 4–8 nm in diameter. Two emission peaks at 320 and 425 nm were observed for the S, N-CQDs. Due to the inner filter interaction, the S, N-CQDs could detect MTX sensitively and selectively. The fluorescence detection method had a linear range of 1–300 μmol/L and an LOD of 0.33 μmol/L (3s/k). Moreover, the S, N-CQDs can be employed to detect MTX in medicine with a high recovery (98–101%). The approach for detecting MTX was simple and rapid, with high selectivity, specificity, and stability, suggesting that S, N-CQDs possess great potential in pharmaceutical analysis.

## Figures and Tables

**Figure 1 molecules-27-02118-f001:**
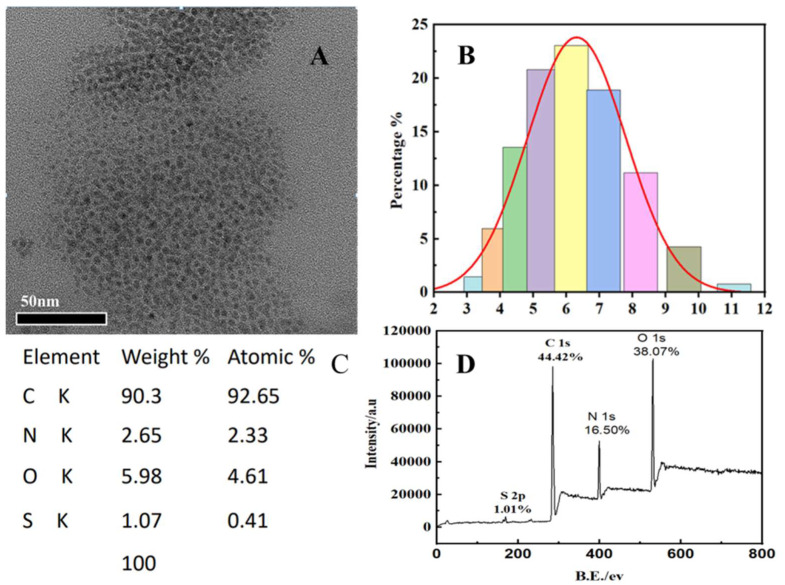
(**A**) HRTEM image of the synthesized S, N-CQDs; (**B**) S, N-CQDs particle size distribution; (**C**) Element composition of S, N-CQDs; (**D**) XPS of S, N-CQDs.

**Figure 2 molecules-27-02118-f002:**
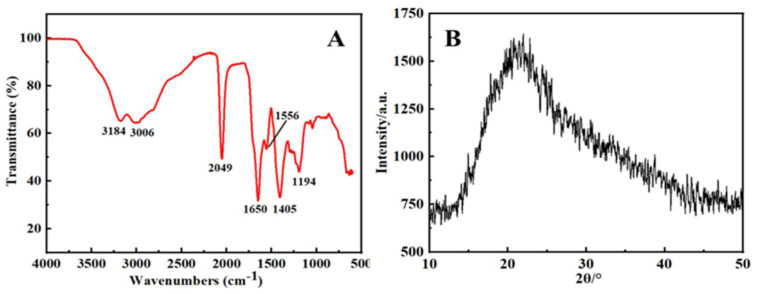
(**A**) FTIR spectrum of S, N-CQDs; (**B**) XRD pattern of S, N-CQDs.

**Figure 3 molecules-27-02118-f003:**
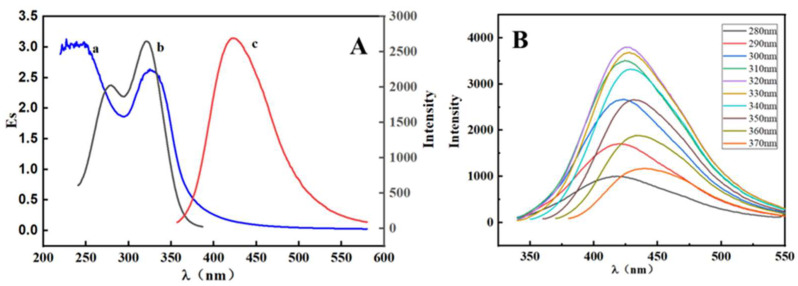
(**A**) UV–Vis absorption (a), excitation (b), and emission (c) of S, N-CQDs; (**B**) Emission spectra at various wavelengths of excitation (280–370 nm).

**Figure 4 molecules-27-02118-f004:**
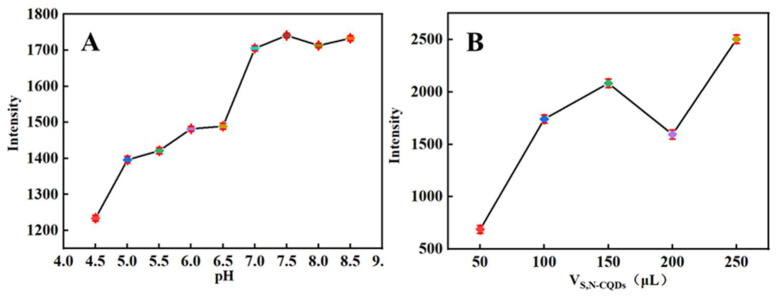
(**A**) Fluorescence response of S, N-CQDs at pH values from 4.5 to 8.5 in the presence of 50 μmol/L MTX; (**B**) fluorescence response of S, N-CQDs at dosage values from 50 to 250 μL in the presence of 50 μmol/L MTX.

**Figure 5 molecules-27-02118-f005:**
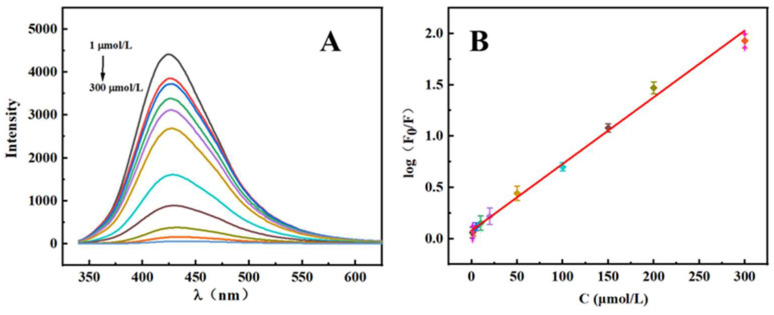
(**A**) Fluorescence intensity curves of MTX with different concentrations; (**B**) linear relation curve of log (F_0_/F) and concentration of MTX.

**Figure 6 molecules-27-02118-f006:**
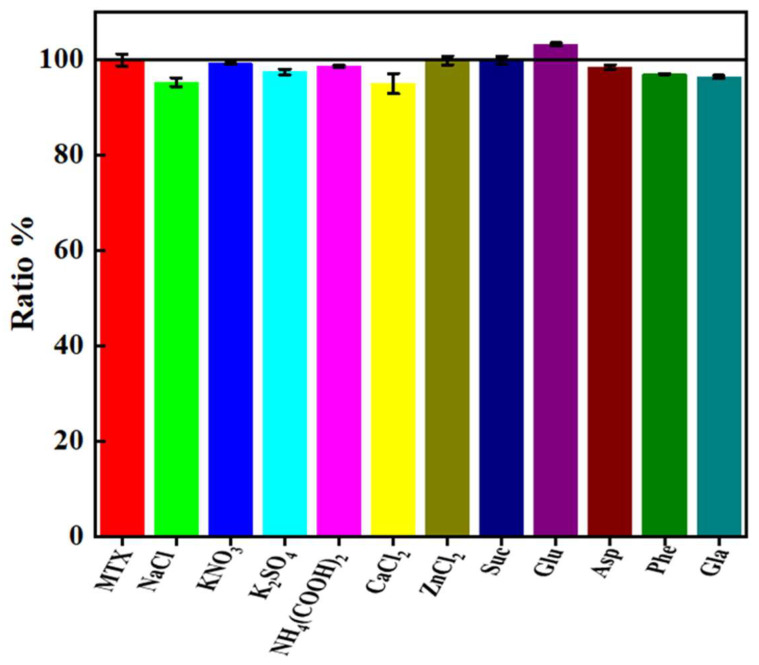
Influence of some metal ions and important biological substances on the 50 μM MTX.

**Table 1 molecules-27-02118-t001:** Comparison of major characteristics for MTX with other published methods.

Method	Linear Range	Detection Limit	References
SPRS	0–150 μmol/L	0.6 μmol/L	[1]
HPLC	5.0–150 μg/mL	3.3 μg/mL	[9]
electroanalysis	0.02–1.5 μmol/L	8 nmol/L	[37]
fluorescence	0.4–41.3 μg/mL	12 ng/mL	[28]
	1–300 μmol/L	0.33 μmol/L	This work

**Table 2 molecules-27-02118-t002:** Determination of MTX concentration in medicine (*n* = 5).

No.	Detected (μmol/L)	Detected (mg/Piece)	Labeled (mg/Piece)	Labeled Quantity (%)	Average (%)	Relative Standard Deviation (%)
1	6.05	2.52	2.5	100.8	99.0	1.1
2	5.93	2.47		98.8		
3	5.92	2.46		98.4		
4	5.90	2.45		98.0		
5	5.95	2.48		99.2		

**Table 3 molecules-27-02118-t003:** MTX standard recovery (*n* = 5).

No.	Sample (μmol/L)	Added (μmol/L)	Total (μmol/L)	Recovery (%)	Average (%)	RSD (%)
1	5.95	80	81.61	94.5	94.9	2.13
2			81.01	93.9		
3			80.87	93.7		
4			80.68	98.5		
5			81.11	94.0		

## Data Availability

The data presented in this study are available on request from the corresponding author. The data are not publicly available due to school rules.

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
