# Peer review of "One-Step Preparation of S, N Co-Doped Carbon Quantum Dots for the Highly Sensitive and Simple Detection of Methotrexate"

_molecules, 2022, doi:10.3390/molecules27072118_

Round 1

Reviewer 1 Report

The manuscript was investigated the preparation of S, N co-doped CQDs for the detection of methotrexate (MTX). The prepared S, N co-doped CQDs were studied using HRTEM, FTIR, XRD, UV Vis absorption. A major comment on this investigation is missing the purpose/ main roles of S and N co-dopants on CQDs. The pure CQDs and single doped on the CQDs should be prepared and act as a control sample for the detection of MTX. Besides, XPS studies should be demonstrated to show the chemical state for the presence of S and N in your prepared CQDs. Introduction is missing the literature review about the common method /materials used for the detection of methotrexate (MTX) before investigating on the CQDs. What is the problem statement and the novelty of this research.

Author Response

Replay: Thank you for your suggestions. The sentences have been added and highlighted in red in the introduction.

Because the atomic radius of N and the electronegativity of S are similar to those of carbon,show an improved electrocatalytic activity and a higher QY, thus paving the way for a new generation of smart electroctalysts, whose activity could be correlated with light emission [22-24]. Compared with single-heteroatom doping, co-doping multiple heteroatoms will greatly improve the performance of CQDs due to the synergistic effect between the doped heteroatoms [25-27]. Moreover, N, S, and C are essential elements in the organism. Therefore, the study of S and N co-doped CQDs (S, N-CQDs) is very meaningful for CQDs in biomolecule and pharmaceutical analysis. However, there are few reports about of S, N-CQDs in the applications for pharmaceutical analysis, and first used to detect MTX [25-28].

Reviewer 2 Report

This manuscript reports the preparation of heteroatom-doped of carbon quantum dots and fluorescence sensing towards methotrexate through a one-stage hydrothermal method, and their corresponding characterizations of structures and sensory properties were also conducted by using FTIR, UV, PL, and TEM. The results of this article present significant scientific contributions with respect to the development of chemosensory materials. I feel that major modifications are necessary before publication can be considered.

1. The biggest concern of the reviewer is the synthetic routes of synthesis part. The authors should provide the synthetic routes and chemical structures of reactants (citric acid and thiourea) and final product. It’s well known that the used reaction conditions (i.e., feed molar ratio, concentration, solvent reaction temperature, reaction time) play a crucial role in determining the size and emission wavelength of carbon quantum dots (QDs). But the authors only used one condition (citric acid (0.003 mmol) and thiourea (0.006 mmol), 180oC, 12 hours, and 30 mL water) for preparing the QDs.

2. The second concern of reviewers is the molecular interaction of S,N-CQD and methotrexate. Is the photoinduced energy transfer or electron transfer for causing the fluorescence quenching of QDs? In reference 28 (Analytica Chimica Acta 1047 (2019) 179), the authors mentioned that the fluorescent probe turn off was originated from the π-π stacking effect and hydrogen interaction. But how about this case?

3. The TEM images (Figure 1A) are not very clear to define the sizes of The DLS data should be given to demonstrate the size distribution of CQDs. In Figure 1A, the authors used the molar ratio 1: 2 of citric acid and thiourea, why the S atom ratio in EDS only have 1.07 wt%?

4. The authors only used the FT-IR spectra to confirm the surface functional groups. However, XPS data is the most convincing evidence available for chemical bonding. Besides, the reviewer wonders that why the S,N-CQD have the O=C=O functional group (Figure 2A)?

5. In the section 3.3.1 (synthesis of S,N-CQD), the size of dialysis membranes should be provided.

6. In the section 2.2, the statement “As shown in a & b of Figure 3(A), the UV absorption spectrum of MTX and the excitation spectrum of the S,N-CQDs exhibit a strong overlap, indicating the presence of an inner filter effect in the current detection system. As a result, the presence of MTX can be detected using a considerable change in fluorescence intensity generated by the inner filter effect” was unclear. The authors should describe it in detail.

7. The authors should compare the sensing ability and LOD of QD with previous methods for the detection of

8. In section 2.2, the authors mentioned that to the best of our understanding, this is a product of S,N-CQDs with a high QY. But the QD in references 22 and 28 revealed higher QY (73 and 57.2%).

9. Many statements should be described in detail.

a. In line 102 of section 2.2, the MTX should be removed.

b. In Figure 4A, the concentration of S,N-CQD should be given.

c. The title of Figure 5A should be corrected.

d. In Table 1, the fully name of RSD should be provided.

Author Response

Comment 1. But the authors only used one condition (citric acid (0.003 mmol) and thiourea (0.006 mmol), 180oC, 12 hours, and 30 mL water) for preparing the QDs.

Reply: Thank you for your suggestions and helpful suggestions. In fact, we have done a lot of work in determining the synthesis scheme of carbon quantum dots. Under the premise of a total volume of 30mL, we synthesized the carbon quantum dots with the molar ratios of citric acid and thiourea at 1:1, 1:2 and 2:1, respectively. In the synthesis of S, N-CQDs, under high temperature and pressure, the amine group in thiourea can react with hydroxyl group in citric acid to form hydrogen bond, the amine group can react with carboxyl group in citric acid to form ionic bond, and the amine group can also react with carbonyl group to form covalent bond. Furthermore, we know that thiourea sublimates at 150-160℃ and decompositions at 180℃ in vacuum. In the temperature optimization experiment, the TEM image of the prepared S, N-CQDs scanned by is not ideal when the temperature is 160℃ or below. When the temperature is high, there may be carbonization to form precipitation, sulfur content is not as much as the ideal, so it is speculated that sulfur dioxide is generated in the reaction process, resulting in low sulfur content. Moreover, the lower the amount of thiourea, the lower the sulfur content, so, 180℃, citric acid and thiourea (1:2) is the best. In the optimization of reaction time, 12h, 16h and 20h were tested, but the TEM images were not very clear, so the reaction time was determined to be 24h. The goal of this manuscript was to share a sensitive and simple method for methotrexate, so the synthesis scheme of S, N-CQDs was not described in detail.

Comment 2. The second concern of reviewers is the molecular interaction of S, N-CQD and methotrexate. Is the photo induced energy transfer or electron transfer for causing the fluorescence quenching of QDs? In reference 28 (Analytica Chimica Acta 1047 (2019) 179), the authors mentioned that the fluorescent probe turn off was originated from the π-π stacking effect and hydrogen interaction. But how about this case?

Reply: Thank you for your comments. The quench fluorescence of S, N-CQD is caused by electron transfer. The mechanism of S, N-CQDs quench fluorescence was added in the 2.2.  After carefully study of the reference you mentioned, we agree with the mechanism of quench fluorescence between S, N-CQDs and MTX. So, the result of this case is consistent with the reference.

Comment 3. The TEM images (Figure 1A) are not very clear to define the sizes of The DLS data should be given to demonstrate the size distribution of CQDs. In Figure 1A, the authors used the molar ratio 1: 2 of citric acid and thiourea, why the S atom ratio in EDS only have 1.07 wt%?

Reply: Thank you for your suggestions. We have also noticed the problem you mentioned in data analysis. It is speculated that sulfur dioxide is generated in the reaction process, resulting in low sulfur content. The sizes of S, N-CQD were determined using the Zetasizer Nano ZS90 (Malvern, UK), as shown as Figure 1. (B).

Comment 4. The authors only used the FT-IR spectra to confirm the surface functional groups. However, XPS data is the most convincing evidence available for chemical bonding. Besides, the reviewer wonders that why the S, N-CQDs have the O=C=O functional group (Figure 2A).

Reply: Thank you for your suggestions. We have taken the XPS experiment of S, N-CQDs. The result was added in Figure 1. (D) and marked in red. On the basis of the experimental results, there is no O=C=O functional group. We have deleted the sentence. Thank you again.

Comment 5. In the section 3.3.1 (synthesis of S, N-CQD), the size of dialysis membranes should be provided.

Reply: Thank you for your comments. The size of dialysis membranes is Mw = 1000 Da. And we have added in 3.3.1 and marked in red.

Comment 6. In the section 2.2, the statement “As shown in a & b of Figure 3(A), the UV absorption spectrum of MTX and the excitation spectrum of the S,N-CQDs exhibit a strong overlap, indicating the presence of an inner filter effect in the current detection system. As a result, the presence of MTX can be detected using a considerable change in fluorescence intensity generated by the inner filter effect” was unclear. The authors should describe it in detail.

Reply: Thank you for your comments. After carefully study of the previous references, the quenching mechanisms for MTX to S, N-CQDs may be related to inner filter effect (IFE) and electron transfer mechanism Because of p-p stacking between the aromatic heterocyclic part of S, N-CQDs and the pteridine ring or phenyl group of MTX. With these specific interactions, N, S co-doped CQDs strongly target methotrexate to serve as chemosensor for the determination of methotrexate based on IFE. We have added in 2.2 and marked in red.

Comment 7. The authors should compare the sensing ability and LOD of QD with previous methods for the detection of MXT.

Reply: Thank you for your comments. Comparison of our method for MTX with other published methods has been added in manuscript (Table 1).

Comment 8. In section 2.2, the authors mentioned that to the best of our understanding, this is a product of S, N-CQDs with a high QY. But the QD in references 22 and 28 revealed higher QY (73 and 57.2%).

Reply: Thank you for your comments. The QY of our experiment is 10.3%. So it is not high. So we deleted the sentence.

Comment 9. Many statements should be described in detail.

Reply: Thank you for your comments. We have revised them and marked in red.

Reviewer 3 Report

see attached file

Author Response

Comment 1: FTIR: maybe the peak at 1650 cm-1 could be the amide C=O stretching band and the ones at 1556 cm-1 and 1405 cm-1 the COO- asymmetric and symmetric stretching of carboxylates, respectively. See also https://doi.org/10.1016/j.electacta.2021.138557

Reply: Thanks for your suggestion. We have revised them and marked in red in 2.1.

Comment 2. Page 3, lines 101-104 the sentence “As 101 shown in a & b of Figure 3(A), the UV absorption spectrum of MTX and the excitation spectrum of the S, N-CQDs exhibit a strong overlap, indicating the presence of an inner filter effect in the current detection system” is misleading as in Fig. 3 the spectrum of MTX is not present…..

Replay: Thank you for your comments. After carefully study of the previous references, the quenching mechanisms for MTX to S, N-CQDs may be related to inner filter effect (IFE) and electron transfer mechanism Because of p-p stacking between the aromatic heterocyclic part of S, N-CQDs and the pteridine ring or phenyl group of MTX. With these specific interactions, N, S co-doped CQDs strongly target methotrexate to serve as chemosensor for the determination of methotrexate based on IFE. We have added in 2.2 and marked in red.

Comment 3. pH-dependence of CQDs PL-emission: the authors chose to perform their MTX detection measurement at pH 6.5 and not at pH 7 where considering Fig. 4A PL-emission of CQDs seems to be more stable….I suggest giving a better discussion also in the light of these works.

Replay: Thank you for your comments. Because the quenching mechanisms for MTX to S, N-CQDs may be related to inner filter effect (IFE) and electron transfer mechanism, so the S, N-CQDs are pH-sensitive with a broad response region is caused by electron transfer.

Comment 4.  Fig. 4B the authors are invited to give a better explanation of the choice of working with 200 ml of CQDs solutions and to give an estimation of CQDs concentration. Replay: Thank you for your comments. I am sorry about it. The citric acid (0.5764 g, 0.003 mmol) and thiourea (0.4567 g, 0.006 mmol) were dissolved in 30 mL ultrapure water. We obtained 25 mL transparent aqueous solution of S, N-CQDs without freeze drying. So, we do not know the concentration of S, N-CQDs. 

Comment 5. Material and Methods section: for sake of clarity the authors are invited to put this section after the introduction or after the conclusions section at the end of the article. Replay: Thank you for your comments. Material and Methods section should be after the introduction. We finished it according to the submission template. If the manuscript is lucky enough to be accepted, we will adjust the structure of the article as required.

Comment 6. Abstract: the authors are invited to not structure the abstract in sub-sections. There are several syntax errors in the manuscript and the authors are advised to reconsider the manuscript.

Replay: We greatly appreciate your comments. We have asked a professor to polish our paper. Please see if the revised version met the English presentation standard.

Round 2

Reviewer 1 Report

The authors addressed the comments accordingly. 

Reviewer 2 Report

The authors have revised the manuscript according to the comments and concerns of the reviewers. The title of Figure 1A should be corrected. And then, this article could be accepted for publication of this journal, Molecules.

Reviewer 3 Report

The authors have significantly improved the manuscript from previous version. The detail statement and corrections make the work acceptable to be published in Molecules. This paper is useful for most researchers working on luminescent nanosized carbon-based materials for sensing applications. I recommend for publication in the present form.